# Sonographic and Magnetic Resonance Characteristics of Gynecological Sarcoma

**DOI:** 10.3390/diagnostics13071223

**Published:** 2023-03-23

**Authors:** Carolina Camponovo, Stephanie Neumann, Livia Zosso, Michael D. Mueller, Luigi Raio

**Affiliations:** 1Department of Obstetrics and Gynecology, University Hospital Insel, University of Bern, 3010 Bern, Switzerland; 2Faculty of Medicine, University of Bern, 3012 Bern, Switzerland

**Keywords:** uterine sarcomas, ultrasound, magnetic resonance imaging, leiomyosarcoma, endometrial stroma sarcoma, STUMP, undifferentiated uterine sarcoma, myomas, carcinosarcoma

## Abstract

Introduction: Gynecological sarcomas are rare malignant tumors with an incidence of 1.5–3/100,000 and are 3–9% of all malignant uterine tumors. The preoperative differentiation between sarcoma and myoma becomes increasingly important with the development of minimally invasive treatments for myomas, as this means undertreatment for sarcoma. There are currently no reliable laboratory tests or imaging-characteristics to detect sarcomas. The objective of this article is to gain an overview of sarcoma US/MRI characteristics and assess their accuracy for preoperative diagnosis. Methods: A systematic literature review was performed and 12 studies on ultrasound and 21 studies on MRI were included. Results: For the ultrasound, these key features were gathered: solid tumor > 8 cm, unsharp borders, heterogeneous echogenicity, no acoustic shadowing, rich vascularization, and cystic changes within. For the MRI, these key features were gathered: irregular borders; heterogeneous; high signal on T2WI intensity; and hemorrhagic and necrotic changes, with central non-enhancement, hyperintensity on DWI, and low values for ADC. Conclusions: These features are supported by the current literature. In retrospective analyses, the ultrasound did not show a sufficient accuracy for diagnosing sarcoma preoperatively and could also not differentiate between the different subtypes. The MRI showed mixed results: various studies achieved high sensitivities in their analysis, when combining multiple characteristics. Overall, these findings need further verification in prospective studies with larger study populations.

## 1. Introduction

Uterine sarcomas are a heterogeneous group of rare malignant tumors with an incidence of 1.5–3/100,000 and are responsible for 3 to 9% of all malignant tumors of the uterus. They originate from the myometrial smooth muscle cells, the endometrial stroma, or the uterine connective tissue. In the groups of malignant mesenchymal tumors (MMT) and malignant mixed epithelial-mesenchymal tumors, the WHO currently differentiates between leiomyosarcoma (LMS), low-grade endometrial stroma sarcoma (LG-ESS), high-grade endometrial stroma sarcoma (HG-ESS), undifferentiated uterine sarcoma (UUS), and adenosarcoma (AS). Earlier classifications also included carcinosarcoma (CS) (also known as malignant mullerian mixed tumors (MMMT)), which is currently classified as a uterine carcinoma. Furthermore, there are smooth muscle tumors of uncertain malignant potential (STUMP), an entity that comprises tumors which cannot be classified as either a malignant leiomyosarcoma or a benign leiomyoma [1]. Leiomyosarcoma are the most common uterine sarcoma, accounting for 60 to 70%, followed by low-grade endometrial stroma sarcoma, high-grade endometrial stroma sarcoma, and undifferentiated uterine sarcoma, with approximately 10% each [1].

With the development of more minimally invasive treatment options for symptomatic myoma, such as uterine artery embolization or intra-abdominal morcellation, there is mounting interest in the preoperative differentiation between uterine sarcoma and benign myoma, as several of these therapeutic options mean undertreatment for sarcoma or risk of intra-abdominal dissemination [2,3,4]. Currently, no clinical symptom, laboratory test, or imaging study can provide effective preoperative diagnostic modalities for uterine sarcomas [5]. The differential diagnosis between myomas and uterine sarcomas is a clinical challenge also because symptoms are often similar; these may include irregular vaginal bleeding, abdominal pain, and a palpable mass [2,6].

Therefore, about 0.1 to 0.3% of patients operated under the diagnosis of uterine leiomyoma are estimated to have a uterine sarcoma [7]. Intraperitoneal morcellation of sarcoma leads to a significantly higher risk of abdominal/pelvic recurrence and a significantly shorter recurrence-free survival; with treatments such as uterine artery embolization there is no histopathological confirmation of the diagnosis at all [8,9]. Hence, the ability to preoperatively differentiate these tumors from usually harmless benign uterine myomas would be of great value for choosing the appropriate therapeutic options.

The ultrasound (US) is the first-line diagnostic tool for gynecological neoplasms, as it is widely available, inexpensive, and not invasive. An often-discussed limitation of the US is that the quality of the examination is dependent on the experience of the examiner and the quality of the device used. The magnetic resonance imaging (MRI) is the preferred imaging technology for preoperative assessment of suspicious uterine lesions, as it is superior in providing morphologic information on the soft tissue intensity to the computed tomography, as this does not offer adequate contrast resolution for delineating focal myometrial masses [6,10]. There are currently no reliable or distinct characteristics for sarcomas defined on any imaging modality, including the MRI and US. Positron emission tomography/computed tomography (PET/CT) has mostly a role in the detection of metastases and recurrence [11].

Laboratory tests are not currently routinely used in the differentiation between uterine sarcomas and myoma, some studies have shown that the levels of serum lactate dehydrogenase (LDH) total activity and its isoenzymes may be relevant in the preoperative diagnosis of uterine sarcomas [12]. The combined use of the dynamic MRI and the serum measurement of LDH (isozymes) seems also to be useful [13,14]. Further studies show that a combination of LDH, D-dimer, and C-reactive protein might be useful for distinguishing uterine sarcomas from the especially degenerated or atypical leiomyoma [15]. In conclusion, the LDH total activity and its isoenzymes may play a role in the preoperative evaluation of the suspect uterine mass. However, further studies are necessary to determine its actual reliability.

Endometrial sampling has a significantly lower predictive value in diagnosing uterine sarcomas compared with epithelial uterine malignancies [16]. In some study results, endometrial biopsy or curettage, which seem to have a similar accuracy, may detect uterine sarcomas in approximately in 25% to 50% of cases [17,18], mostly in cases of endometrial involvement by the tumor. Therefore, a negative biopsy does not preclude the diagnosis of uterine sarcoma, until the complete hysterectomy specimen is examined [5].

An ultrasound-guided trans-uterine cavity core needle biopsy of uterine myometrial tumors is technically possible and could confirm or exclude a possible malignant tumor. As it requires an invasive procedure and an experienced examiner, the core-needle biopsy is currently not part of the guidelines of the preoperative workup. The overall diagnostic accuracy is described between 93% and 98%, although data on patients with sarcoma is very rare [19].

Due to the lack of specific symptoms, laboratory testing, and preoperative histological sampling, which accurately detect the cases of uterine sarcomas, and because of the relevance of the preoperative differentiation of both entities to assess the advisable operative procedure, we decided to focus on the characteristics of the imaging currently described in the literature.

The objective of this review is to gain an overview of the studies published in the last 10 years proposing characteristic of the different sarcoma subtypes on the ultrasound and MRI, respectively, as well as of the studies assessing the accuracy of the same imaging techniques for detecting gynecological sarcoma preoperatively.

## 2. Materials and Methods

As a first step in developing the search strategy, the research question was split up into individual concepts using the PICO approach. For each concept both MeSh and Emtree terms, respectively, and text words were identified. The actual search was then conducted in Medline (Ovid) and EMBASE (Ovid). The searches were separated for the MRI and ultrasound, and for both imaging technologies the search was conducted in Medline and Embase.

We decided to include CS and STUMP in the results because the older categorizations often applied in the reviewed timespan still included CS as a sarcoma subtype and some studies did not differentiate between STUMP and LMS. We also decided to limit the search to studies published in the last ten years to ensure the currently available technology would be reviewed, especially considering retrospective studies usually use data going back additional years.

For the ultrasound, the search in Medline was conducted on the 7 October 2021 and Embase was searched on the 13 October 2021 using MeSh and Emtree terms and text words to search the title and abstract for the concepts “ultrasonography” and “sarcoma” or “endometrial stromal tumors” or “leiomyosarcoma”, and in Embase the term “female genital system” was added to get more precise results. Additional limits were female and human, and the year of publishing was from 2011 to current. The language was also limited to English, German, French, or Italian. This yielded a total of 424 results in Medline and another 552 results in Embase, so 976 records in total.

The search for the results on the MRI was conducted analogously to the one for the ultrasound. Medline and EMBASE were searched on the 14 October 2021. The same limits were applied. These searches yield 197 results in Medline and 660 results in EMBASE, therefore, a total of 857 records. All search strategies can be found in detail in the appendix.

The title and abstract of the search results were reviewed in search of the relevant information, and the following were excluded: case reports, conference abstracts, publications on non-female-genital sarcoma (soft tissue, breast) or on other gynecological neoplasms; and publications with a focus on therapy using ultrasound instead of diagnostics, with a focus on clinical characteristics instead of imaging, with a focus on artificial intelligence or machine learning, with a focus on pediatrics, or with a focus solely on the comparison of different imaging techniques instead of the tumor characteristics. As an additional criterion, the studies had to include at least six patients with sarcoma. Duplicates in the results of the subsequently performed searches in Embase were excluded manually. 

For the ultrasound, 15 reports were retrieved, and 23 reports were retrieved for the MRI. The PRISMA flow charts in “Figure 1” and “Figure 2”, respectively, illustrate the steps of the report selection.

The retrieved studies were then read, and data was extracted on the study characteristics and results using Microsoft Excel (Microsoft Corporation). The relevant study characteristics included the type of study, year of publication, sarcoma subtype, number of patients, and precise question answered in the study. Ultimately, 12 studies on the ultrasound and 21 studies on the MRI were included in the review. 

## 3. Results

### 3.1. Ultrasound

#### 3.1.1. General Characteristics of Sarcoma

In the following paragraphs, the general characteristics of sarcoma on the ultrasound are mentioned and the corresponding study are listed in Table 1. Kohler et al. [20] found a suspicious sonography to be a significant difference between leiomyoma (10%) and leiomyosarcoma (81%) (*p* < 0.001). A “suspicious sonography” was defined if any of the following findings were present: (1) poorly defined borders to the myometrium; (2) tumor having mostly heterogenous echogenicity with large areas of strong hyperechogenicity; (3) patchy or predominately hypo-to-anechoic regions across the entire tumor; (4) borders between areas of different echogenicity being derounded (often bizarre or with pointed, tapered extensions); or (5) serosa reached or ruptured.

Cho et al. [9] observed that if the largest-diameter lesion was larger than 8 cm, sarcoma was the more likely diagnosis than myoma (*p* = 0.006), with a hazard ratio of 3.584.

Ludovisi et al. [7] analyzed the ultrasound characteristics of uterine sarcoma of 116 patients with leiomyosarcoma, as well as 48 with endometrial stromal sarcoma and 31 with undifferentiated endometrial sarcoma in a large retrospective multicenter study. Most cases were large, solid tumors with a median diameter of 91 mm and with inhomogeneous echogenicity; about half also had cystic areas. Other prevalent findings were irregular tumor borders and moderate or rich vascularization on the color Doppler, as well as the absence of shadows and absence of calcifications.

Gao et al. [21] retrospectively analyzed 80 cases of uterine sarcoma in China (from 1988 to 2007). The following ultrasound characteristics were observed: in all cases the size of the uterus was enlarged, in part with unclear capsular boundaries or irregular forms. The internal of the mass was, for the most part, uneven, sometimes also a liquid dark area or low heterogeneous echo. The mean tumor diameter was 8.23 cm. The uterine sarcoma showed an abundant blood stream, and, especially in the periphery of the tumor, blood flow signal indicating neovascularization was present.

Cheng et al. [22] compiled common imaging findings in 72 patients with uterine sarcoma. The ultrasound showed rich blood flow in 33.4% of the patients, no clear margin in 20.8%, mass degeneration in 31.9%, and none of the above manifestations in 13.9%.

Bonneau et al. [23] compared the ultrasound findings of 23 patients with MMT and STUMP to those of benign leiomyomas. The significant findings were that MMT/STUMP presented more often as a single mass with an OR of 6.9 (95% CI 2.2–22.4), more frequently had a non-myometrial origin with an OR of 12 (95% CI 3.4–41.9), a thickened endometrium or an intra-cavitary process with an OR of 13.9 (95% CI 3.8–50.8), and no acoustic shadowing with an OR of 11.8 (95% CI 1.7–84.5). In cases where sonographers described the difficulties in characterizing a pelvic mass, MMT/STUMP were discovered significantly more frequently than benign leiomyomas (*p* = 0.001).

#### 3.1.2. Characteristics of Sarcoma Subtypes

Some studies also analyzed the features of specific subtypes of sarcoma. Alcazar et al. [24] described the ultrasound features of uncommon primary malignant ovarian tumors, including nine sarcomas (four carcinosarcoma, three fibrosarcoma, one liposarcoma, one sarcoma). These tumors mostly presented as large unilateral solid masses with moderate or abundant vascularization.

Ciccarone et al. [25] collected the ultrasound characteristics of ovarian carcinosarcomas in a retrospective multicenter study with 91 patients. All tumors contained solid components, most (73%) were purely solid, and on the color Doppler most (85%) were moderately or richly vascularized. In all cases, the solid component was inhomogeneous with irregular margins. The largest tumor diameter was in the median 100 mm. Acoustic shadowing was only present in one case. The most common ultrasound appearance was a large solid tumor containing cystic areas with inhomogeneous echogenicity of the solid tissue and irregular tumor borders, followed by a large multilocular-solid mass with inhomogeneous echogenicity of the solid tissue.

Park et al. [26] retrospectively evaluated the ultrasonographic findings in ten cases of low-grade endometrial stromal sarcoma. All tumors were located intramurally, some protruding into the endometrial cavity. Some masses were well-defined, others ill-defined, and some showed diffuse myometrial thickening. Multiseptated cystic degeneration or multiple small areas of cystic degeneration are common findings. Based on the findings, the cases were categorized into four patterns: (1) a predominantly solid mass containing cystic areas (six cases), (2) a predominantly unilocular cystic mass (one case), (3) an ill-defined infiltrative solid mass mimicking adenomyosis (two cases), and (4) a well-defined solid mass (one case).

Ludovisi et al. [7] also highlighted some differences in the ultrasound features between the different subtypes of sarcoma. Leiomyosarcomas were larger than ESS and UUS. Endometrial stromal sarcomas showed the highest percentage of visible normal endometrium (91.7%) and regular tumor borders (60.4%), and had less vascularization than other sarcomas. Undifferentiated uterine sarcomas had the highest rate of irregular tumor borders (74.2%), absent shadowing (87.1%), and hemorrhagic or ground-glass echogenicity of cyst fluid (40.0%).

#### 3.1.3. Accuracy

A few studies also evaluated the diagnostic accuracy of ultrasonography to detect sarcoma or predict malignancy in a pelvic mass. Li et al. [27] conducted a multicenter, retrospective study on patients with uterine sarcoma in Western China, including an examination of the accuracy of the preoperative diagnosis. The diagnostic sensitivity of the ultrasound for uterine malignant tumors was 11.0%, meaning only in 11 out of 100 patients was the preoperative finding on the ultrasound a malignant tumor. In an additional 33% malignancy was suspected, which should be understood as an ambiguous conclusion. The remaining 56% were falsely categorized as benign tumors on the ultrasound. The ultrasound achieved the highest diagnostic sensitivity in the USS group (33.3%), but the difference was not statistically significant.

Najibi et al. [28] compared the diagnostic accuracy of contrast-enhanced/DWI MRI and ultrasonography in the differentiation between benign and malignant myometrial tumors. The cross-sectional study observed 63 patients that underwent surgery for intrauterine masses and were assessed using the ultrasound and MRI. On the ultrasound, sarcoma was diagnosed in 25.4% of cases, while after the surgery, sarcoma was diagnosed histologically in 58.7% of cases or 37 patients. This led to a sensitivity of 35.1%, specificity of 88.4%, PPV of 81.2%, NPV of 48.9%, and accuracy of 57.1%, respectively. In the chi-square test for the association of pathological results with ultrasonography and MRI findings, the US performed considerably lower than the MRI (*p* = 0.034 vs. 0.0001).

Gaetke-Udager et al. [29] examined the diagnostic accuracy of the ultrasound for differentiating leiomyosarcoma from leiomyoma. In addition, five-point Likert scores were assigned for the following features: margins, necrosis, hemorrhage, vascularity, calcifications, heterogeneity, and likelihood of malignancy. Based on these scores, the overall suspicion scores for malignancy were calculated and the receiver operating characteristic curves were generated. The receiver operating characteristic (ROC) curves for the discrimination of leiomyosarcoma from leiomyoma were not significantly different from chance for the ultrasound.

**Table 1 diagnostics-13-01223-t001:** Overview of reviewed studies on ultrasound.

Author	Year	Study Type	Number of Patients	Sarcoma-Subtypes (Number of Patients)	Objective
Kohler et al. [20]	2019	Prospective study	293	LMS	Developing a preoperative leiomyoma score
Gaetke-Udager et al. [29]	2016	Retrospective study	10	LMS	Diagnostic accuracy of ultrasound for LMS vs. LM
Cho et al. [9]	2016	Retrospective study	31	14 ESS, 11 LMS, 6 US	Identify preoperative diagnostic findings suggestive of uterine sarcoma
Ludovisi et al. [7]	2019	Retrospective multicenter study	195	116 LMS, 48 ESS, 31 UES	Clinical and ultrasound characteristics of uterine sarcomas
Gao et al. [21]	2014	Retrospective study	80	38 ESS, 22 LMS, 18 CS, 2 US	Characteristics of uterine sarcoma (in China)
Li et al. [27]	2020	Retrospective study	114	50 LG-ESS, 34 LMS, 13 HG-ESS, 9 UUS, 8 AS	The accuracy of preoperative diagnosis with US
Alcazar et al. [24]	2012	Retrospective study	9	4 CS, 5 others	Gray-scale and color Doppler ultrasound features of uncommon primary malignant ovarian tumors
Cheng et al. [22]	2020	Retrospective study	72	27 ESS, 20 LMS, 15 AS	Common imaging findings of uterine sarcoma
Najibi et al. [28]	2021	Cross-sectional study	37	not specified	Diagnostic accuracy of ultrasound in benign vs. malignant myometrial tumors
Ciccarone et al. [25]	2021	Retrospective multicenter study	91	CS (Ovaries)	Clinical and ultrasound characteristics of ovarian carcinosarcoma
Park et al. [26]	2016	Retrospective analysis	10	LG-ESS	US findings associated with LG-ESS
Bonneau et al. [23]	2013	Retrospective cohort study	23	7 UES, 6 CS, 4 STUMP, 3 LMS, 2 LG-ESS	US performance for differentiating LM vs. MMT

### 3.2. MRI

Some studies did not differentiate between the different subtypes of sarcoma when evaluating the characteristics. Those results will be listed first, followed by the studies focusing on certain subtypes, as well as special imaging techniques, and lastly the studies assessing the accuracy. The corresponding studies are listed in Table 2. The typical MRI findings in the imaging of malignant tumors include that the high signal intensity (SI) on T1-weighted imaging (T1WI) can be a sign of hemorrhage or of methemoglobin, more precisely. The high signal intensity on T2-weighted imaging (T2WI) can typically be caused by necrosis. Both hemorrhage and necrosis also cause heterogeneous SI on the MRI, as well as heterogeneous enhancement on the contrast-enhanced MRI (CE-MRI). Hyperenhancement on CE-MRI can be a sign of neovascularization represented by the contrast agent leakage. The diffusion-weighted imaging (DWI) characterizes tissue based on the diffusion motion of the water molecules. It allows quantitative measurements in the form of the apparent diffusion coefficient (ADC) values. In tumors with a high nuclear-to-cytoplasm ratio, the diffusion is limited, which is expressed by lower ADC values [10,30,31,32].

#### 3.2.1. Characteristics for All Subtypes

Thomassin-Naggara et al. [33] found the following significant characteristics for predicting malignancy: intermediate T2-weighted signal intensity, high *b*_1000_ SI on DWI, lower mean ADC values, heterogeneity on T2WI, intratumoral hemorrhage, endometrium thickened or not seen, and heterogenous enhancement, as well as patient age. A recursive partitioning model using high *b*_1000_ SI, T2WI SI, and mean ADC achieved 92.4% accuracy. 

Bi et al. [34] found ill-defined tumor margins, solid parts with hyperintense or mixed signal on T2WI, and solid parts being hypointense on ADC maps signifying restricted diffusion, as well as low values on ADC overall to be significant parameters to distinguish uterine sarcoma from atypical leiomyoma. Abnormal vaginal bleeding, ill-defined tumor margins, location mainly in the uterine cavity, and mean ADC below the cutoff combined in the predictive score achieved a sensitivity of 88.9% and a specificity of 99.9%.

Takeuchi et al. [35] evaluated the option of using susceptibility-weighted MR (SWI) sequences for diagnosing uterine sarcomas. SWI is particularly sensitive to distortions of the local magnetic field and can be used to detect, for example, blood products [36]. Using T2 star-weighted MR angiography (SWAN), signal voids signifying intratumoral hemorrhages were detected in all cases of sarcoma. The accuracy, sensitivity, and specificity for SWAN were 97%, 100%, and 96%. Additional findings include all sarcomas showing heterogeneous high signal intensity on DWI and T2WI. An ADC value below the cutoff resulted in a sensitivity of 91.7%, specificity of 90%, and accuracy of 91.2%. The ADC values differed significantly between sarcoma and leiomyoma, but there was an overlap in the individual values. 

Sumi et al. [37] used quantitative assessments of MR images to find differences between the major histological types of uterine sarcomas, as well as between the malignant and benign tumors. The focus was on the contrast ratio (CR) of the signal intensity in T2WI for the tumor areas, compared with the iliopsoas muscle, and on the contrast-enhanced ratio (CER) of different parts of the tumor after gadolinium enhancement on T1WI, with the latter used to quantify the heterogeneity. These ratios were calculated for leiomyosarcoma, carcinosarcoma, and endometrial stromal sarcoma, as well as leiomyoma. The different entities were characterized as follows: ESS have solid parts showing homogenous gadolinium enhancement in younger patients; LMS have areas of lower SI on T2WI in larger myometrial tumors with irregular contours and hemorrhage; and LM have more homogenous gadolinium enhancement and lower SI on T2WI. Additionally, significant differences in qualitative assessments favoring uterine sarcoma were the irregular contours and the presence of hemorrhagic, necrotic, and cystic components. 

Malek et al. [38] investigated quantitative parameters based on T2WI and contrast-enhanced MRI with the psoas muscle and outer myometrium as internal references to calculate the various ratios. Uterine sarcomas consistently scored significantly different values for all quantitative metrics from benign myoma; some measurements achieved a sensitivity for sarcoma of 100% with a specificity of 89%. Furthermore, malignant lesions showed a high SI on T2WI, and central necrosis was ten times more common in sarcoma than myoma. 

#### 3.2.2. Characteristics Only LMS 

Sahin et al. [39] assessed the ability of non-contrast MRI features to differentiate uterine leiomyosarcoma from atypical benign leiomyomas. The most prevalent features in the leiomyosarcoma group were an at least intermediate T2 signal intensity of the solid areas (93.8%) and cystic or necrotic alterations (93.8%). A significant difference (*p* ≤ 0.001) between LMS and leiomyoma was found for intratumoral hemorrhage, interruption of the endometrial interface, irregular tumor shape, and thickened or not seen endometrial stripe, with all being more present in LMS. The highest odds ratios for the prediction of LMS was for the irregular tumor shape and interruption of endometrial interface (12.00 and 64.00, respectively). All ADC measurements were consistently lower for leiomyosarcoma, but the statistical significance was not reached.

Li et al. [40] investigated DWI for differentiating uterine LMS from degenerated leiomyoma. LMS had statistically significant more often ill circumscribed margins, a more hyperintense SI on DWI, and lower ADC values.

Lakhman et al. [8] assessed the use of qualitative MRI features to distinguish the LMS form atypical leiomyoma (ALM). Four features were most significantly associated with LMS: nodular borders, intra-lesional hemorrhage, “T2 dark” areas, and central unenhanced areas. When three out of four of these features were present, the highest combined sensitivities and specificities for LMS were found (both 0.95–1.00).

The two following studies limited leiomyomas in the control group to tumors that showed a high signal intensity on T2WI or T1WI, respectively. Both are characteristics that are usually attributed to malignant tumors. Rio et al. [41] compared the MRI features of leiomyosarcoma with the atypical and degenerated leiomyomas that also showed hyperintensity on T2WI. The following features were found significantly more often in LMS: irregular borders, “T2 dark” areas, presence of central necrosis, presence of high signal on b1000 DWI, ADC value below cutoff, and hyperenhancement of the tumor, compared to the myometrium on post-contrast images. Predictive of the malignancy was the presence of irregular borders and central necrosis.

Ando et al. [42] compared the characteristics of leiomyosarcoma and leiomyomas with hyperintense areas on T1WI (T1 HIA). Leiomyosarcomas were more heterogeneous, ill-demarcated, and had a higher occupying rate and a lower signal intensity ratio. Leiomyomas more frequently showed a T2 hypointense rim within the T1 HIA and a higher signal intensity ratio of T1 HIA on T1WI.

#### 3.2.3. Characteristics Other Subtypes of Sarcoma

A few studies also focused on more rare subtypes of gynecological sarcoma. Saida et al. [43] compared the features of carcinosarcoma of the ovary with high grade serous carcinoma (HGSC). CS was statistically significantly larger, and the stained-glass appearance of cystic components, hemorrhage, and necrosis were also significantly more common in CS of the ovary.

Li et al. [44] investigated features in 15 cases of endometrial stromal sarcoma on conventional MRI and DWI. Almost all cases were solid tumors with heterogeneous hyperintense signal intensities on T2WI, obvious enhancement on contrast-enhanced MRI, and hyperintensity on DWI, clearly depicting the border of the tumors. Most tumors showed cystic degenerations, necrosis, and/or hemorrhage. Additionally, the ADC value was found to inversely correlate with the Ki-67 expression.

Huang et al. [45] compared characteristics of high- and low-grade ESS on DWI and CE-MRI. HG-ESS showed more feather-like enhancement, hemorrhage, and necrosis. The feather-like enhancement achieved an accuracy of 95% in the differentiation of HG and LG-ESS. Comparing ESS with leiomyoma, LM had higher ADC values and showed no ill-defined margins, worm-like nodules, or feather-like enhancement, unlike ESS. Additionally, ESS showed much more necrosis and hemorrhage. It was concluded that DWI is the preferred imaging modality to differentiate between ESS and LM, while CE-MRI is superior when differentiating between LG- and HG-ESS.

Bi et al. [46] compared the MRI characteristics of the different sarcoma subtypes. Carcinosarcoma were mostly endometrioid shaped and most had cystic changes or necrosis, which appeared mostly patchy. LMS predominantly showed slit-like cystic changes or necrosis. ESS more commonly showed a band sign on T2WI. On T1WI, LMS and ESS were isointense compared to CS being slightly hypointense. The solid components of LMS showed mixed signals on T2WI, compared to the hyperintense signals found in CS. 

Zhang et al. [47] assessed the conventional MRI and DWI for the categorization of uterine sarcoma subtypes. No significant differences between the subtypes were found, except for a more heterogeneous SI on T2WI of LMS, compared to ESS and CS. However, a significant divergence of ADC values between LMS and LM was found, with the values for LMS lying below the cut off.

### 3.3. Special Imaging Techniques

Lakhman et al. [8] further used the 16 qualitative MRI features they found differentiating LMS from ALM to evaluate the feasibility of texture analysis. Combined, these 16 findings suggested a greater textural heterogeneity of LMS. They deduced that the texture analysis was a feasible approach for semi-automated lesion categorization. Texture analysis is an image analysis technique allowing for more elaborate lesion characterization than the regular MRI sequences that are visually and subjectively analyzed by radiologists. It is a semi-automated method based on quantitative metrics and can be used to assess lesion heterogeneity and the presence of sub elements within lesions [48].

Gerges et al. [48] performed texture analysis of multiple MRI sequences and assessed its utility for differentiating LMS and LM. Texture analyses using metric obtained from T2-weighted images performed best at differentiating LMS from LM; the highest-achieved sensitivity and specificity being 82.4% and 74.5%.

Rahimifar et al. [49] evaluated ^1^H magnetic resonance spectroscopy (MRS), and compared and combined it with DWI to classify the tumors as malignant or benign. MRS detects markers of biochemical processes and can provide information on metabolism, transformation into malignant tissue, and presence of active tumors. In their study, DWI restrictions, MRS choline peak, and MRS lipid peak were all significantly more often found in malignant mesenchymal tumors. There was a significant difference in the mean ADC of malignant tumors to benign ones, but there was an overlap in the range of the values. The highest specificity, NPV, and accuracy (100%, 100%, 98.3%) was found for the combination of an ADC below the cutoff value and MRS positive for the lipid or choline peak.

#### Accuracy

In a previously mentioned study, Najibi et al. [28] compared the diagnostic accuracy of contrast-enhanced/DWI MRI and ultrasonography. In contrast to ultrasonography, with the MRI, the correct diagnosis was made in all but four cases, leading to a sensitivity of 94.6%, specificity of 92.3%, positive predictive value of 94.6%, negative predictive value of 92.3%, and accuracy of 93.7% for the MRI to detect sarcoma. These values were not affected by the baseline clinical conditions, such as abnormal uterine bleeding, pain severity, or even menopausal status.

Gaetke-Udager et al. [29] also examined the diagnostic accuracy of the conventional MRI for differentiating leiomyosarcoma from leiomyoma. Again, five-point Likert scores were assigned for the same qualitative features as on the ultrasound and ROC curves were calculated for assessing the ability to predict malignancy. The results did not vary from chance for the MRI without DWI.

Li et al. [27] found in their retrospective examination of the accuracy of preoperative diagnosis using MRI in 34 patients with uterine sarcoma a sensitivity of only 35.3%. In an additional six patients, malignancy was suspected, but roughly half of the patients received a benign preoperative diagnosis.

Lakhman et al. [8] found that if at least three out of four of the following MR features—nodular borders, intra-lesional hemorrhage, “T2 dark” areas, and central unenhanced areas—were present, LMS could be diagnosed with combined sensitivities and specificities of 0.95–1.00 each.

Thomassin-Naggara et al. [33] developed a recursive partitioning model using high *b*_1000_ signal intensity, T2WI signal intensity, and mean ADC value, which achieved an accuracy of 92.4%.

Malek et al. [38] evaluated the quantitative parameters, based on T2WI and contrast-enhanced MRI. The classifiers combining the tumor myometrium contrast ration on CE-MRI and T2WI yielded the highest sensitivity and specificity at 100% and 89%.

Bi et al. [34] proposed a preoperative predictive score combining abnormal vaginal bleeding, ill-defined tumor margins, location mainly in the uterine cavity, and mean ADC below cutoff. The achieved values for the sensitivity were 88.9%, the specificity 99.9%, the accuracy 95.7%, and the positive and negative predictive values were 97.0%, and 95.1%.

Takeuchi et al. [35] assessed susceptibility-weighted MR sequences for diagnosing uterine sarcoma and found an accuracy, sensitivity, and specificity of 97%, 100%, and 96%. The positive predictive value, negative predictive value, positive likelihood ratio, and negative likelihood ratio were 91%, 100%, 25, and 0. Using a cutoff ADC value of 0.97 × 10^−3^ mm^2^/s for uterine sarcomas resulted in a sensitivity of 91.7%, specificity of 90%, and accuracy of 91.2%.

Lin et al. [50] compared the diagnostic accuracy of the contrast-enhanced MRI and diffusion-weighted MRI for differentiating uterine leiomyosarcoma/STUMP from benign leiomyoma. The CE-MRI achieved a significantly superior diagnostic accuracy (0.94 vs. 0.52) and a significantly higher specificity (0.96 vs. 0.36) than DWI (*p* < 0.05 for both), while keeping a similarly high sensitivity (0.88 vs. 1.00). The combination of DWI and an ADC value below the cutoff yielded a comparably high diagnostic performance as CE-MRI, with an accuracy, sensitivity, and specificity of 0.88 each.

Rahimifar et al. [49] evaluated MR spectroscopy for diagnosing uterine sarcoma. The combination of an ADC below the cutoff value and MRS positive for lipid or choline peak achieved the highest specificity, NPV, and accuracy (100%, 100%, 98.3%).

**Table 2 diagnostics-13-01223-t002:** Overview of reviewed studies on MRI.

Author	Year	Study-Type	Number of Patients	Sarcoma-Subtypes (Number of Patients)	Objective
Li et al. [27]	2020	Retrospective study	34	15 LG-ESS, 10 LMS, 5 HG-ESS, 3 UUS, 1 AS	The accuracy of preoperative diagnosis with MRI
Sumi et al. [37]	2015	Retrospective study	25	11 CS, 8 LMS, 6 ESS	Distinguish major histological types of uterine sarcomas
Saida et al. [43]	2021	Retrospective case-control study	12	CS (ovary)	Imaging and clinical characteristics of ovarian carcinosarcoma (CS) compared with high-grade serous carcinoma.
Takeuchi et al. [35]	2019	Retrospective case-control study	10	6 CS, 3 LMS, 1 ESS	Susceptibility-weighted MR sequences (SWS) for diagnosis of sarcoma
Lin et al. [50]	2015	Prospective study	8	6 LMS, 2 STUMP	Diagnostic accuracy of CE-MRI vs. DWI for LMS/STUMP vs. LM
Sahin et al. [39]	2021	Retrospective case-control study	16	LMS	Non-contrast MRI features of LMS and atypical LM
Rahimifar et al. [49]	2019	Prospective study	14	Not specified	DWI and MR-Spectroscopy for differentiation; combining ADC and MRS for better accuracy
Lakhman et al. [8]	2017	Retrospective study	19	LMS	Qualitative MR features to distinguish LMS from ALM, feasibility of texture analysis
Li et al. [40]	2017	Retrospective study	16	LMS	DWI for differentiation LMS and degenerated LM
Li et al. [44]	2017	Retrospective study	15	13 LG-ESS, 2 HG-ESS	Conventional MRI and DWI features of ESS and correlation of ADC-value and Ki-67 expression
Gerges et al. [48]	2018	Retrospective study	17	LMS	Texture analysis of multiple MRI sequences for differentiation of LMS and LM
Thomassin-Naggara et al. [33]	2013	Retrospective study	25	9 UES, 4 CS, 3 LMS, 2 LG-ESS, 1 RMS, 6 STUMP	MRI for differentiation malignant vs. benign
Malek et al. [38]	2019	Prospective study	14	Not specified	Diagnostic accuracy of preoperative quantitative metrics based on T2WI and CE-MRI
Zhang et al. [47]	2014	Prospective study	22	7 LMS, 9 ESS+AS, 6 CS	MRI and DWI for categorization of uterine sarcoma (compared to pathology)
Rio et al. [41]	2019	Retrospective study	20	LMS	MRI features differentiating atypical and degenerated LM with hyperintensity on T2WI from LMS
Bi et al. [46]	2020	Observational study	71	29 ESS, 27 CS, 15 LMS	MRI features incl. ADC for preoperative identification of sarcoma subtypes
Ando et al. [42]	2018	Retrospective study	19	14 LMS, 5 STUMP	Differences of LMS vs. LM with T1WI hyperintense areas (T1HIAs)
Bi et al. [34]	2018	Retrospective study	36	24 ESS, 12 LMS	Qualitative and quantitative MRI features of sarcoma vs. ALM
Huang et al. [45]	2019	Retrospective study	20	11 HG-ESS, 9 LG-ESS	Diagnostic accuracy of MRI in diagnosing and differentiating HG- vs. LG-ESS
Najibi et al. [28]	2021	Cross-sectional study	63	Not specified	Diagnostic accuracy CE/DWI-MRI for differentiating malignant vs. benign myometrial tumors
Gaetke-Udager et al. [29]	2016	Retrospective study	7	LMS	Diagnostic accuracy MRI without DWI for LMS vs. LM

## 4. Discussion

### 4.1. Ultrasound

Six studies [7,9,20,21,22,23] evaluated the characteristics on the ultrasound that can help to diagnose gynecological sarcoma and differentiate them from leiomyoma. Across these studies, the following features, listed in Table 3, were reported multiple times. The features common in gynecological sarcoma were a large, solid tumor with a diameter >8 cm and poorly defined borders. The tumors had heterogeneous echogenicity and produced no acoustic shadowing. Furthermore, moderate or rich vascularization and blood flow were reported and often cystic changes or degenerations within the tumor were found. These findings are supported by the current literature. Dueholm and Hjorth [51] summarized mostly the same imaging features when listing the imaging features that raise suspicion of leiomyosarcomas (in their review on diagnosis of abnormal uterine bleeding). Additional features proposed are a single lesion and a lack of calcification. Poorly defined borders are a sign of the infiltrative nature of sarcoma. The rapid growth of sarcomatous tumors may exceed their blood supply, leading to avascular areas of either hemorrhagic or cystic degeneration, which present as heterogeneous lesions with irregular hypoechoic to anechoic areas [52].

One study evaluates ten cases of low-grade ESS [26] and describes four patterns. Two studies focus on features of ovarian carcinosarcoma and common features of both include the tumors being solid or predominantly solid and having moderate or abundant vascularization. Currently, hardly any literature exists on the ultrasound patters of LG-ESS and CS, so these findings cannot be further verified. Oh et al. [3] also mention in their review on ultrasound features of uterine sarcomas that there is a wide overlap of practically all of the features attributed to the various subtypes, and many are also not specific for malignancy. Because of the lack of verified imaging appearance of sarcoma subtypes, there is currently no application for the ultrasound in the differentiation between subtypes of gynecological sarcoma. 

In a retrospective study, Bonneau et al. [23] found that, if the sonographers described difficulties in characterizing a pelvic mass, MMT/STUMP were significantly more frequently diagnosed than benign leiomyomas. This may be in part explained by the fact that typical myoma have a common appearance of a round shape with sharp margins, and a mostly homogenous echotexture. Significant vascularization on the color Doppler imaging is an unusual finding. Larger leiomyomas may appear more heterogeneous on the US due to areas of degeneration [52]. This makes degenerated myomas more difficult to differentiate form sarcoma because they have a more similar heterogeneous echogenicity and central necrosis are also possible in cases of atypical benign lesions [4].

This may be a reason for the following results. Two studies evaluated the preoperative diagnostic accuracy of the ultrasound for sarcoma, but sensitivities of only 11% [27] and 35% [28], respectively, were reached. Gaetke-Udager et al. [29] examined the diagnostic accuracy further using ultrasound features that match the above discussed features compiled in this review. The ROC curves for differentiating sarcoma from leiomyoma did not significantly differ from chance. Sun et al. [4] mention in their review that increased vascularity on the color Doppler ultrasound, combined with a large size and degenerative cystic changes, indicate malignancy with a sensitivity of 75% and PPV of 60%. There might be some value in using the ultrasound to identify gynecological sarcoma. A newer technique mentioned in the literature, that was not assessed in the reviewed studies, is the three-dimensional (3D) power Doppler angiography. It allows a more objective assessment of the tumor volume and vascularization using calculations of the vascularity index, flow index, and vascularity-flow index, and might be used in the future to differentiate benign from malignant tumors [2,10].

**Table 3 diagnostics-13-01223-t003:** Key features of sarcoma on ultrasound.

Key Ultrasound Features of Sarcoma	
Tumor size	Large, diameter >8 cm
Type of tumor	Solid
Borders	Poorly defined
Echogenicity	Heterogeneous
Shadowing	No acoustic shadowing
Vascularization	Moderate to rich vascularization
Degenerations	Cystic changes or degenerations common

### 4.2. MRI

The results were split up in multiple sections, gathering studies with similar focuses and they will now be discussed the same way. Across the studies not differentiating sarcoma between subtypes [33,34,35,37,38], the following features, listed in Table 4, were found multiple times: sarcomas showed heterogeneous and intermediate to high signal intensities on T2WI. These findings are consistent with the presence of intratumoral hemorrhage and necrosis [32]. Intratumoral hemorrhage and necrosis, as well as cystic degeneration, are also explicitly noted as features. These are common histopathological findings in gynecological sarcomas [1]. Another frequent feature is irregular or ill-defined borders. Lastly, a well-documented finding is hyperintense SI on DWI and low ADC/below the cutoff, which are also associated with intratumoral hemorrhage and necrosis [30]. Huang et al. [30] mention all features listed above in their review on the current status of MRI in malignant uterine neoplasm. Smith et al. [31] developed an aide-memoire with the name BET^1^T^2^ER Check! to help identify the features suggestive of uterine sarcoma, with each letter representing a feature. Irregular borders is listed as a key suspicious feature for sarcoma’ T2 signal intensity is described as intermediate and heterogeneous, depending on the areas of necrosis and the hemosiderin deposits caused by hemorrhage. Restricted diffusion manifesting as high SI on DWI and low ADC values is also a common feature. The remaining features of the acronym BET^1^T^2^ER were heterogeneous enhancement on CE-MRI, low signal intensity on T1WI with high signal intensity in areas of the hemorrhage, and endometrial thickening, all of which are also mentioned in some of the reviewed studies but not as consistently as the features discussed above. 

Another issue that was noted by Smith et al. [31], as well as several of the reviewed studies, was an overlap in ADC values between sarcoma and atypical leiomyomas. Typical leiomyomas are easily distinguished from sarcoma by their appearance as well-circumscribed, round, or oval shape, and with homogenous low SI on T2WI relative to the myometrium, no hyperintensity on DWI, and higher ADC values ranging from 1.2 to 1.7 × 10^−3^ mm^2^/s, as well as isointensity on T1WI. Atypical or degenerated leiomyomas, on the other hand, show variable appearances on the MRI, some overlapping with those attributed to sarcoma. Cystic and myxoid degenerations are hyperintense on T2WI; red degenerations caused by hemorrhagic infarction show heterogeneous, high SI on T1- and T2WI [2]. Cellular myomas are mildly hyperintense on T2WI due to their high cellularity, which also shows as restricted diffusion presenting as high SI on DWI and low ADC values [31].

These issues were taken into account in most studies [8,39,40,41,42] examining the differentiation of leiomyosarcoma and myoma by choosing only atypical or degenerated LM to compare to the cases of LMS. The features noted above for sarcoma in general are also verified in these studies on LMS specifically: irregular borders, heterogeneous, high signal intensity on T2WI, cystic, hemorrhagic, and necrotic changes especially central necrosis presenting as central non-enhancement, as well as hyperintensity on DWI and low values for ADC. These findings are consistent with the current literature [4,32]. Another feature mentioned in two studies [8,41] is the “T2 dark” area. This represents areas of previous hemorrhage that have a signal intensity lower than the muscles on T2WI [32]. A high SI on T2WI is a marker for tissue cellularity; in case of leiomyosarcoma, it is a result of cellular atypia and high mitotic rates, which are typical histopathological findings in LMS. High SI on DWI illustrate restricted diffusion, which is caused by the high cellular density of LMS representing the nucleus-to-cytoplasm ratio [4]. Virarkar et al. [53] performed a meta-analysis with eight studies, some of which were also reviewed here, on the diagnostic performance of high SI on T1- and T2WI and ADC values for differentiating uterine LMS from benign leiomyoma. They found that high SI on T1WI and low ADC can differentiate LMS from LM. On T2WI, LMS had higher pooled sensitivities, but the result did not reach statistical significance. The signal intensity on T1WI was not a relevant feature in the reviewed studies, but, in general, it can be said that the sarcoma can have a variable SI on T1WI. Frequently, there are some areas of higher SI caused by necrosis or hemorrhage. 

Some studies also investigated the features of more rare subtypes of sarcoma. In the literature, carcinosarcoma are described as an endometrial mass with low to isointense SI to the myometrium on T1WI, and heterogeneous hyperintensity on T2WI, suggestive of intratumoral necrosis or hemorrhage [30]. The findings in the reviewed studies [43,46,47] agree with this description. Endometrial stromal sarcoma is also a focus of a few reviewed studies [44,45,46,47]. They are described as solid tumors with heterogeneous hyperintensity on T2WI and DWI, and obvious enhancement on CE-MRI. Most tumors showed hemorrhage and necrosis; these features were even more prevalent in HG-ESS. Feather-like enhancement was also typical for HG-ESS. Worm-like nodules were another feature for ESS. These findings are consistent with the current literature [30,31]. The wormlike nodules are a typical feature for the growth of LG-ESS along lymphatic and vascular vessels. Feather-like enhancement is the most accurate characteristic for HG-ESS, and consists of a fine, wispy enhancement scattered within tumor cells because of myometrial invasion. 

The same studies that retrospectively reviewed the accuracy of the ultrasound in diagnosing gynecological sarcoma also evaluated the MRI. Li et al. [27] evaluated 34 patients with sarcoma, for which a sensitivity of only 35% was reached; roughly half of the patients received a benign preoperative diagnosis. The study of Najibi et al. [28] had more promising results with a correct diagnosis of sarcoma made in 94% of the cases. On the other hand, an evaluation of the qualitative features on conventional MRI by Gaetke-Udager et al. [29] could not achieve an accuracy that differed from chance. These are only three studies, but they highlight the uncertainty of the MRI for a definitive diagnosis of sarcoma, preoperatively. Several authors proposed a combination of characteristics for the improved accuracy for the diagnosis of uterine sarcoma. Good results were achieved with the following approaches: contrast-enhanced MRI alone, as well as with a combination of DWI features and ADC value below cutoff [50]; a combination of at least three out of four of nodular border, intra-lesional hemorrhage, “T2 dark” areas and central unenhanced areas [8]; high *b*_1000_ signal intensity, T2WI signal intensity, and mean ADC value [33]; Tumor myometrium contrast ratio on CE-MRI and T2WI [38]; as well as abnormal vaginal bleeding, ill-defined border, location in uterine cavity, and mean ADC below cutoff [34]. This seems like a promising approach, as most of these features can be evaluated with the standard MRI sequences for the suspected gynecological sarcoma, but there is certainly further evaluation needed.

A few studies explored the imaging modalities outside of the standard MRI protocols. Two studies [37,38] calculated quantitative ratios based on T2WI and CE-MRI that compared sarcomas to the psoas muscle or myometrium. Some metrics were able to identify sarcoma with a sensitivity of 100% and a sensitivity of 89%. The use of quantitative measures is a relevant addition to the current potentially interobserver variable qualitative assessment. Takeuchi et al. [35] evaluated the option of using susceptibility-weighted MR sequences for diagnosing uterine sarcomas. Using T2 star-weighted MR angiography signal voids signifying intratumoral hemorrhage were detected in all cases of sarcoma. The accuracy, sensitivity, and specificity reached were 97%, 100%, and 96%. SWI is a relative new modality and is currently mainly used in neuroimaging to identify hemorrhage or calcium [36], but its ability to identify small amounts of hemorrhage is certainly of interest in the examination of potential sarcoma. In a recent study, Rahimifar et al. [49] evaluated a different approach using MR-spectroscopy to classify tumors as malignant or benign. Specifically, the choline and lipid peaks on MRS were found as significant features for sarcoma. Combined with the ADC values below the cutoff, they reached an accuracy of 98%. The limitations of MRS are the additional acquisition time necessary for post processing and difficulty in shimming, due to the air and intestinal movement, so it is currently not widely applied for diagnosing gynecological sarcoma [30]. Lakhman et al. [8] evaluated the feasibility of texture analysis as a quantitative, semi-automatic evaluation of MRI sequences and found a greater textural heterogeneity in LMS. Gerges et al. [48] later performed TA on multiple MRI sequences and found that metrics obtained from T2WI were suited best for differentiating LMS from LM, with a maximum sensitivity of 82.4%. The application of texture analysis is currently just a research technique, but these results show the potential of a computer analysis of already standard MRI sequences to improve diagnostic accuracy.

**Table 4 diagnostics-13-01223-t004:** Key features of sarcoma on MRI.

Key MRI Features of Sarcoma	
Borders	Irregular or ill-defined
SI on T2WI	Heterogeneous and intermediate to high SI
Degeneration	Hemorrhage, necrosis, cystic degenerations
SI on DWI	Hyperintense SI
ADC value	Low/below cutoff
Enhancement	Heterogeneous enhancement on CE-MRI
SI on T1WI	Low SI with areas of high SI

### 4.3. Other Imagings

Fluorine-18-fluorodeoxyglucose positron emission tomography (FDG-PET) has been used for the diagnosis of various types of malignant tumors. Numerous studies discussed its usefulness in uterine sarcomas; however, most of them reported on the usefulness of FDG-PET in the diagnosis of the recurrence of disease. Nagamatsu et al. [54] analyzed the value of FDG-PET, combined with the serum lactate dehydrogenase (LDH) levels and compared with FDG-PET alone for the diagnosis of leiomyosarcomas (LMS). FDG-PET imaging of endometrial cancer (EC) was used as a reference. The study reported that the intratumoral standardized uptake value (SUV) obtained from FDG-PET imaging was useful for the differential diagnosis of leiomyosarcoma and leiomyoma, and that when the cut-off for SUV was set to 3, the rate of diagnostic accuracy was 0.79 (sensitivity, 1.0; specificity, 0.73). Future studies are needed to determine if FDG-PET could have a role in the preoperative prediction of uterine sarcomas.

### 4.4. Laboratory Testing

Elevated lactate dehydrogenase (LDH) seems to be a useful biomarkers for the preoperative diagnosis of LMS [14].

Goto et al. conducted a prospective study to identify the magnetic resonance imaging characteristics of uterine leiomyosarcoma (LMS) and to evaluate the diagnostic accuracy of conventional MRI and dynamic MRI by Gd-DTPA, with or without serum measurement of lactate dehydrogenase (LDH) levels. The specificity, positive predictive value, negative predictive value, and diagnostic accuracy were 93.1%, 52.6%, 100%, and 93.1% with MRI alone; 93.8%, 83.3%, 100%, and 95.2% with dynamic MRI alone; and 100%, 100%, 100%, and 100% with combined use of LDH and MRI, respectively.

The combined use of dynamic MRI and serum measurement of LDH (isozymes) seems to be useful in making a differentiated diagnosis of LMS from degenerated leiomyomas, but cellular leiomyoma and STUMP were often candidates for false-positive results in this study [14].

Nagai et al. presented a retrospective study with 63 patients, with 15 diagnosed with uterine sarcoma and 48 with benign tumors, and developed the PREoperative sarcoma score (PRESS score), which was intended to serve as reference for selecting therapeutic strategies. The authors concluded that the LDH values, and MRI and endometrial cytology findings are significant predictors of uterine sarcoma in both groups; PRESS was considered a prototype diagnostic score and further studies adding new parameters and a bigger population are required [13].

Abnormal endometrial cytology with the presence of spindle or multinucleated giant cells with scanty cytoplasm; relatively large, hyperchromatic nuclei; and conspicuous nucleoli may also be seen if uterine sarcoma protrudes into the uterine cavity. However, these findings are not sufficiently accurate for making a preoperative diagnosis of uterine sarcoma; hysterectomy and histopathological examination are necessary in order to differentiate the uterine sarcoma from uterine myoma, because endometrial cytology or biopsy requires mostly a tumor growth into the uterine cavity [5,17,18,55].

## 5. Conclusions and Limitations

A big limitation of this review is the size of the study populations. For the ultrasound, there are some recent large multicenter studies with over one hundred patients each, but for the MRI especially, many studies have under twenty patients or tumors with sarcoma included, particularly if separated by subtype. In addition, almost all studies were conducted retrospectively. This is of course due to the rarity of sarcomas with an incidence of 1.5–3/100,000 women and could not be rectified by the expansive timespan of the reviewed cases of often over ten years. Still, some skepticism toward the generalizability of the results is warranted. Another limitation is the restriction to publication of the last ten years, respectively, since 2011. Since most publications highlight a slightly new angle on the issue of diagnosing gynecological sarcoma preoperatively, some approaches that were the focused on in older publications might have been missed. This might also explain some minor discrepancies or gaps in the results, compared to the current literature. An additional aspect that was not explicitly assessed in the reviewed studies is the effect of the experience of the observer, i.e., the person conducting the ultrasound or the radiologist assessing the MRI, on the results or accuracy. Regarding this issue, more objective imaging analysis, such as 3D power Doppler angiography for the ultrasound and texture analysis for the MRI, could prove to be valuable additions.

Nevertheless, an overview of the characteristics identifying sarcoma on the ultrasound could be achieved. For the MRI, several studies achieved excellent accuracies with various features and combinations of features for differentiating sarcoma in that setting. These results now need to be further verified, ideally in prospective studies with larger cohorts. This way, hopefully in the future, the consensus on the ultrasound and MRI features significant in the differentiation of gynecological sarcomas and benign leiomyomas could be improved.

## 6. Patents

### Declaration of Authorship

We herewith confirm that we wrote this review without external help and that we did not use any resources other than those indicated.

We have clearly acknowledged all parts of the text where material from other sources has been used, either verbatim or paraphrased.

## Figures and Tables

**Figure 1 diagnostics-13-01223-f001:**
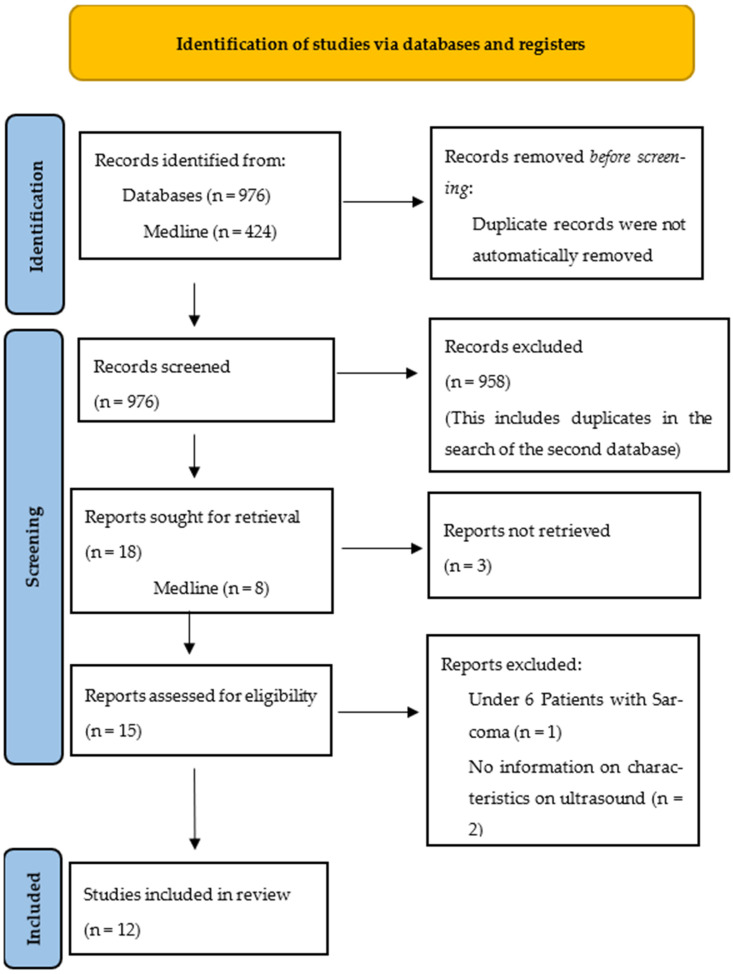
PRISMA flow chart of search on ultrasound.

**Figure 2 diagnostics-13-01223-f002:**
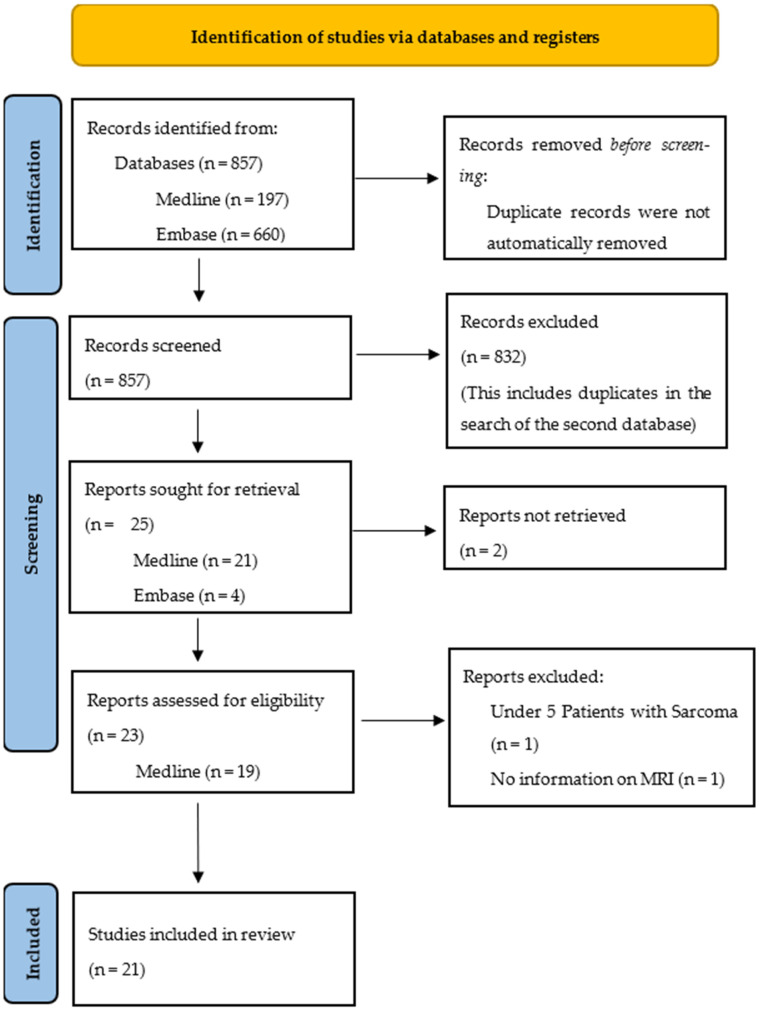
PRISMA flow chart of search on MRI.

## Data Availability

No new data were created, the research of data was conducted in Medline (Ovid) and EMBASE (Ovid).

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
