# Peer review of "Sonographic and Magnetic Resonance Characteristics of Gynecological Sarcoma"

_diagnostics, 2023, doi:10.3390/diagnostics13071223_

Round 1

Reviewer 1 Report

Title:  The title is informative

Abstract

Remove discussion in the section of abstract and substitute it by conclusion

Introduction:

Add clear importance of this systemic review at the end of introduction section before the aim

Methodologies and discussion

classifications of results and methodology are acceptable

Try to cite some references from the diagnostics/MDPI

Figures qualities are good

Conclusions

In my opinion the manuscript as whole is -Informative, but still require editing, and revision as mentioned above in details. With my report and other reviewer/s report the manuscript will meet the publication criteria due to its importance in the field.

Edit some typos

Good luck

Author Response

Thank you very much for your kind review.

  • Abstract: we changed the discussion to conclusion
  • Introduction: we added a section about the importance of our review
  • We added an interesting article from MDPI Diagnostics about preoperative core-needle biopsy
  • We corrected a few typos

Reviewer 2 Report

The submitted manuscript addresses an issue of crucial clinical importance. This work has many strengths and a few weaknesses that can be easily improved. The preoperative differentiation between uterine myomas and sarcomas remains the Achilles heel of the modern gynecology.

This review paper, critically addressing two main imaging modalities, will be of interest to a wide readership. However, imaging is not the only diagnostic method, and ultrasound/MRI are not used in isolation from clinical signs and symptoms, laboratory methods (LDH and its fractions), preoperative histological sampling, etc. Therefore, the introduction and discussion should definitely be enriched with information that embed imaging (and its limitations) in the overall diagnostic process. Please see and discuss  e.g.  Kostov et al. 2021 doi: 10.3390/clinpract11040103 ; Pritts et al. 2015, doi:10.1007/s10397-015-0894-4; Goto et al. 2002 doi:10.1046/j.1525-1438.2002.01086.x, Bansal et al. 2008, doi:10.1016/j.ygyno.2008.02.026

Author Response

Thank you for your kind review.

We added and completed our review with the references you sent us. So that laboratory testing and endometrial biopsy are also mentioned.